# Post-9/11 Mental Health Comorbidity Predicts Self-Reported Confusion or Memory Loss in World Trade Center Health Registry Enrollees

**DOI:** 10.3390/ijerph17197330

**Published:** 2020-10-08

**Authors:** Howard E. Alper, Rifat A. Tuly, Kacie Seil, Jennifer Brite

**Affiliations:** 1New York City Department of Health and Mental Hygiene, Long Island City, NY 11101, USA; kseil@health.nyc.gov (K.S.); jbrite@health.nyc.gov (J.B.); 2School of Public Health, Columbia University, New York, NY 10032, USA; rtuly101@gmail.com

**Keywords:** World Trade Center, 9/11, disaster, mental health, confusion, memory loss

## Abstract

Numerous studies report elevated levels of chronic mental health conditions in those exposed to the World Trade Center attacks of 11 September 2001 (9/11), but few studies have examined the incidence of confusion or memory loss (CML) or its association with mental health in 9/11 attack survivors. We investigated the incidence of CML and its association with the number of post-9/11 mental health conditions (PTSD, depression, and anxiety) in 10,766 World Trade Center Health Registry (Registry) enrollees aged 35–64 at the time of the wave 4 survey (2015–2016) that completed all four-wave surveys and met the study inclusion criteria. We employed log-binomial regression to evaluate the associations between CML and the number of mental health conditions. A total of 20.2% of enrollees in the sample reported CML, and there was a dose-response relationship between CML and the number of mental health conditions (one condition: RR = 1.85, 95% CI (1.65, 2.09); two conditions: RR = 2.13, 95% CI (1.85, 2.45); three conditions: RR = 2.51, 95% CI (2.17, 2.91)). Survivors may be experiencing confusion or memory loss partly due to the mental health consequences of the 9/11 attacks. Clinicians treating patients with mental health conditions should be aware of potential cognitive impairment.

## 1. Introduction

The terrorist attacks on the World Trade Center (WTC) in New York City on 11 September 2001 (9/11), resulted in over 2700 deaths and many thousands more injured. Many survivors were enveloped in the dust/debris cloud generated by the collapse of the WTC towers. Some survivors also witnessed traumatic events such as seeing airplanes strike the towers. These exposures led to the development of a variety of physical and mental conditions, such as asthma [1,2,3,4,5], post-traumatic stress disorder (PTSD) [6,7,8,9], heart disease [6,10,11], stroke [8], and cancer [12,13]. Some of these diseases appeared in survivors in the first several years after the attacks (e.g., asthma, PTSD), while other diseases have longer latency periods and have appeared 10–15 years after the 9/11 attacks (e.g., heart disease, stroke, certain cancers).

As the cohort ages, medical conditions associated with older age will become more common. Cognitive impairment may affect the survivors as they age, so it is important to investigate the incidence of this condition and whether it is associated with conditions that developed following exposure to the 9/11 attacks. For example, a recent publication [14] demonstrated that cognitive impairment was associated with education, social support, and physical activity. A similar result was found by in the “Nun study” [15,16], a non-9/11-related longitudinal study of aging and Alzheimer’s disease in 678 Catholic sisters, which found that multilingualism was associated with a decreased likelihood of developing dementia [17].

Other non-9/11-related research has also shown that cognitive impairment is associated with PTSD [16,18,19,20,21,22,23,24,25,26,27], depression [28,29,30,31,32], general anxiety [33], and comorbid PTSD and anxiety or PTSD and depression [26,34].

Research on 9/11-related risk factors for cognitive impairment in the survivor and responder populations has appeared in recent years. For example, research by Clouston et al. [35] on 9/11 responders has demonstrated elevated levels of mild cognitive impairment (MCI) compared to a control group of age-matched clinical trial participants, and that MCI was associated with PTSD symptom severity, even after controlling for comorbid depression and anxiety. A study of firefighters by Singh et al. [36] demonstrated substantial association between exposure to 9/11 and cognitive function (as measured by the self-report Cognitive Function Instrument (CFI) instrument), which was mediated substantially by PTSD and depression.

While the above studies demonstrate that cognitive impairment is associated with individual mental health conditions, little research has explored how mental health comorbidity, defined here as the number of mental health conditions affecting an individual (PTSD, depression, and generalized anxiety disorder), is associated with cognitive impairment. Furthermore, while much previous 9/11 research on mental health and cognitive impairment has focused on police, firefighters, or related groups, little such research has investigated survivors (residents, area workers, passersby, and local school staff and students). Finally, the present study represents the largest cohort, to our knowledge, that has been used to examine the association between mental health conditions and cognitive impairment.

In this study, we investigate the following: (1) the incidence of CML, (2) the association between the number of mental health conditions and CML, and (3) the association between number of mental conditions and severity of CML, among those enrollees that reported CML.

## 2. Materials and Methods

### 2.1. Data Source and Study Design

The World Trade Center Health Registry (Registry) was established in 2002 to monitor the physical and mental health outcomes among those exposed to the terrorist attacks of 11 September 2001. Enrollees included rescue/recovery workers, residents, area workers, passersby, and students and staff of local schools. The Registry conducted its major surveys in 2003–2004 (wave 1), 2006–2007 (wave 2), 2011–2012 (wave 3), and 2015–2016 (wave 4). The methods used by the Registry have been described in detail in previous publications [3,37]. The Registry protocol was approved by the institutional review boards of the Centers for Disease Control and Prevention and New York City Department of Health and Mental Hygiene, protocol number 02–058. At enrollment, participants in the Registry provided consent and are updated on a periodic basis on their human subject rights.

### 2.2. Analytic Sample

The study inclusion criteria were as follows: (1) enrollees who completed surveys for waves 1–4; (2) enrollees aged 35–64 at wave 4; and (3) enrollees who reported no confusion or memory loss in the twelve months prior to wave 3. Study exclusion criteria were: (1) enrollees who reported having a stroke during any of the wave surveys [35]; and (2) enrollees who had pre-9/11 PTSD, depression, or anxiety. The final analytic sample consisted of 10,766 enrollees (Figure 1).

### 2.3. Study Variables

#### 2.3.1. Outcome Measure

The main outcome was self-reported CML, as defined by the response (yes/no) to the wave 4 survey question, “During the last 12 months, have you experienced confusion or memory loss, other than occasionally forgetting the name of someone you recently met?” A second outcome was the severity of CML classified among enrollees who answered “yes” to experiencing CML, as defined by the survey question, “During the last 12 months, has your confusion or memory loss happened more often or gotten worse?”

#### 2.3.2. Exposure

The primary exposure was mental health comorbidity, defined as the number of mental health conditions reported by the enrollee during the wave 3 survey. Mental health conditions included probable PTSD, depression, and generalized anxiety disorder. Probable PTSD was assessed at wave 3 using the 9/11-specific PTSD Checklist-Specific (PCL-S), a 17-item self-reported symptom scale that specifically references the events of 9/11. Each item is self-rated, based on the previous thirty days, and on a five-point scale from not at all (1) to extremely (5), leading to a total score between 17 and 85. Probable PTSD was defined as a PCL score ≥ 44. The PCL-S is a well-validated measure and has good temporal stability, internal consistency (>0.75), test-retest reliability (0.66), and high convergent validity (0.58–0.93) (Wilkins, Lang, and Norman, 2011), with overall diagnostic efficiency = 0.90, sensitivity = 0.94, and specificity = 0.86 (Blanchard, Jones-Alexander, Buckley, and Forneris, 1996). Probable depression was assessed at wave 3 using the Patient Health Questionnaire (PHQ-8) [38]. The PHQ-8 consists of eight questions that inquire about the frequency of symptoms over the last two weeks, which are each rated from 0 (not at all) to 3 (nearly every day). The total score ranges from 0–24, and a cutoff of 10 was found to possess a sensitivity of 0.70 and specificity of 0.98 [38]. Probable generalized anxiety disorder (GAD) was assessed using the GAD-7 instrument [39], a 7-item scale. Each item is self-rated, based on the previous two weeks, and on a four-point scale from not at all (0) to nearly every day (3), leading to a total score between 0 and 21. A total score of 10 or greater indicates probable GAD. This cutoff leads to a sensitivity of 0.88 and specificity of 0.82. The number of mental health conditions (i.e., the mental health comorbidity) reported at wave 3 could thus be 0, 1, 2, or 3.

#### 2.3.3. Covariates

Covariates included the following sociodemographic characteristics: gender (male, female), age at wave 3 (25–44, 45–64), race/ethnicity (white non-Hispanic, black non-Hispanic, Hispanic, Asian non-Hispanic, other), educational attainment (less than high school/high school diploma/GED, some college, bachelor’s or post-graduate degree), marital status at wave 3 (married/living together, divorced/separated, widowed, never married), social support at wave 3, and rescue/recovery workers status (RRW) at wave 1. Social support was defined by the sum of the answers to five questions, each of which can range from 0 (none of the time) to 4 (all of the time): “How often is someone available to take you to the doctor if you need to go?,” “To have a good time with you?,” “To hug you?,” “To prepare your meals if you are unable to do it yourself?,” and “To understand your problems?” The total score can range from 0 to 20, and low social support was defined as a total score below 15. RRW status was treated as binary (yes (RRW)/no (resident, area worker, passersby, school students/staff)).

### 2.4. Statistical Analysis

The frequency distributions for the outcome (CML) and the main exposure (mental health comorbidity), as well as the sociodemographic variables and protective/risk factors were calculated.

Log-binomial regressions were performed to describe the association between CML and mental health comorbidity, controlling for sociodemographic factors and other covariates. We chose this method because exploratory analyses showed that the incidence of CML was sufficiently large that the odds ratios produced by logistic regression would poorly approximate the risk ratio (RR). Three bivariate regressions of CML on PTSD, depression, and anxiety were performed to determine if the associations of CML with these mental health conditions were similar. This would justify using the sum of these mental health conditions in further analyses. Two multivariable regressions were performed: (1) CML was regressed on mental health comorbidity for the entire analytic sample, and (2) CML severity was regressed on mental health comorbidity for those enrollees who answered “yes” to having CML at wave 4. We first ran regression (1) including an interaction term between mental health comorbidity and RRW status, to determine if the main association of interest differed by the RRW status. If this interaction term was found to be non-statistically significant, it was excluded from regression (1). All calculations were performed using SAS version 9.4 (SAS, Inc., Cary, NC, USA) and SAS-compatible IVEware 0.3 (Survey Methodology Program, University of Michigan, Ann Arbor, MI, USA).

In performing regression (1) above, we found that about 16% of the sample had missing data and were excluded from the model. We tested the effect of missing data by performing multiple imputation (MI) with 10 replicates and using regression (1) to compute average risk ratios for the association of mental health comorbidity and CML, controlling for covariates. The risk ratios using imputation differed little from the unimputed results, so we report the latter in this paper.

## 3. Results

The characteristics of the study sample (*n* = 10,766) are shown in Table 1. While 90.0% of the sample had none of the three mental health conditions included in this study, 6.0% had one, and 4.0% had two or more. The sample was 65.1% male, 68.4% aged 45–64 at wave 3, 73.7% white non-Hispanic, 58.4% having a bachelor’s degree or higher, and 74.8% married/living with partner. Two-thirds (66.5%) reported a high level of social support at wave 3. A total of 50.3% were RRW.

The incidence of CML was 20.2%. The CML incidence for those with zero, one, two, or three mental health conditions at wave 3 was 18.0%, 36.0%, 42.2%, and 50.5%, respectively.

The bivariate regressions of CML on PTSD, depression, and anxiety produced risk ratios (RR) of 2.32, 2.43, and 2.06, respectively. For the main multivariable log-binomial regression, the mental health comorbidity-RRW interaction term was not statistically significant (*p* > 0.29), so this term was excluded from the main regression, and an RRW-only term was included. Results for the main multivariable log-binomial regression are presented in Table 2. CML exhibited a dose-response relationship with mental health comorbidity (one condition: risk ratio (RR) = 1.85, 95% confidence interval (CI): 1.65–2.09; two conditions: RR = 2.13, 95% CI: 1.85–2.45; three conditions: RR = 2.51, 95% CI: 2.17–2.91; dose-response test: *p* < 0.001).

Finally, the regression of worsening CML on mental health comorbidity among enrollees reporting CML at wave 4 (Table 3) showed a modest to strong association (one condition RR = 1.24, 95% CI: 1.01–1.52; two conditions RR = 1.89, 95% CI: 1.57–2.28; three conditions RR = 1.97, 95% CI: 1.62–2.41).

## 4. Discussion

We found that the incidence of confusion or memory loss in our sample was 20.2%. CML exhibited a dose-response relationship with mental health comorbidity. Finally, we showed that worsening CML exhibited a moderate to strong association with mental health comorbidity among enrollees who reported CML.

The present study is consistent with previous research on the association of PTSD and other mental health conditions with mild cognitive impairment. PTSD has been associated with declines in memory and learning [26,27,35], and prospective studies have provided evidence that PTSD is associated with increased risk of developing cognitive decline [22,25]. Further, depression and general anxiety, separately and together, have also been linked to the development of early mild cognitive impairment [34]. Our study is also consistent with recent research on the predictors of mild cognitive impairment in victims of the 9/11 attacks. In particular, Clouston et al. found that PTSD severity was associated with increased likelihood of mild cognitive impairment [26,35]. 

It is important to elucidate the mechanisms relating mental health conditions to cognitive impairment. One well-researched mechanism concerns the effect of stress on hippocampal volume [40,41]. For example, stress can activate the amygdala, which, through its projections to the hippocampus, can lead to reduced hippocampal volume, which is associated with decreased memory function [42]. A second stress-based pathway goes through the hypothalamic-pituitary-adrenal (HPA) axis. Activation of the HPA axis by mental health conditions or stressful life events can damage the neocortex, leading to decrements in multiple cognitive functions [43]. Another mechanism concerns the propensity of mental health conditions to adversely affect physical health in ways that increase the likelihood of cognitive impairment. For example, PTSD and other mental health conditions have been associated with subsequent cardiovascular and cerebrovascular diseases, such as heart attack [44] and stroke [8] (though all enrollees who reported history of stroke in waves 1–4 were excluded in the present study), which increases the likelihood of cognitive impairment. PTSD has also been shown to be associated with the subsequent development of diabetes [45], itself a risk factor for cognitive impairment. Additionally, recent research has also shown that PTSD-induced sleep disturbances may mediate the pathway from mental health disease to cognitive impairment [46]. Finally, hearing loss has been found to be associated with subsequent mental health problems such as depression [47], which can increase the chance of later developing cognitive impairment.

The public health implications of the present study are several-fold. In our study sample, 10% of enrollees had at least one of the specified mental health conditions at wave 3. If the sample had included people 65 years or older and people who had reported CML at W3, those proportions could be even greater. If the results were extrapolated to the entire population of people directly exposed to 9/11, the number of individuals with mental health conditions that could lead to a greater risk of developing CML is substantial. This highlights the need to continue to monitor 9/11-exposed enrollees for the development of mental health conditions.

This study also found that more than one-fifth of enrollees in the study sample suffered from confusion or memory loss at the wave 4 survey and that the probability of CML increased with the number of mental health conditions. This implies that a substantial number of people exposed to the 9/11 attacks could suffer from confusion or memory loss. The likely increasing prevalence of this condition as the 9/11 disaster-exposed population ages highlights the need to monitor 9/11-exposed patients for the emergence of CML, MCI, and dementia, especially among enrollees with mental health conditions.

A major strength of this study is that we used a prospective cohort, which allowed us to investigate the relationship between mental health conditions that developed after the 9/11 attacks but before the self-reported confusion or memory loss. Further, the relatively large sample size gave us enough statistical power to detect associations of interest.

An important limitation is that all data used in this study, including variables for confusion or memory loss, were self-reported. The validity of self-reported measures of cognitive function has been questioned. However, our sample was chosen to encompass a middle-aged group of enrollees, so those in our sample were unlikely to suffer from dementia, which could lead to misclassification in reporting of CML. Further, a recent study of the association between 9/11 exposure and 9/11-related PTSD and changes in self-reported cognitive function using the Cognitive Function Instrument (CFI) [48] demonstrated results that were comparable with those from a recent study that used objective measures of cognitive function [35]. Furthermore, social desirability bias could lead to under-reporting of CML; similar under-reporting could occur for mental health conditions and symptoms. A related limitation is that the question used as the main outcome measure did not allow determination of the severity of cognitive impairment enrollees experienced.

Another limitation arises due to the difficulty in isolating the effect of a single exposure—9/11—on cognitive impairment. Many factors can influence cognitive function. However, we excluded enrollees with pre-9/11 PTSD, depression, and anxiety from our analytic sample, and we included a rich set of covariates.

Since the three mental health conditions investigated in this study were self-reported, they do not necessarily constitute confirmed diagnoses for PTSD, depression, or anxiety. Additionally, our definition of mental health comorbidity was based on the above three conditions, so when an enrollee is classified as having zero conditions, they may in fact have a mental health condition we did not include in our classification. Finally, research has demonstrated a degree of neural overlap between anxiety and mood disorders [49], so the constructs for PTSD, depression, and GAD may not be entirely separable. Their sum may not accurately represent enrollee’s mental health comorbidity.

Another limitation concerns the attrition that has occurred as successive waves of Registry data have been collected. This attrition could introduce bias into estimates of incidence of CML or its association with mental health comorbidity; however, a study by Yu et al. [49,50] found the effects of such attrition on associations of interest to be minimal.

## 5. Conclusions

The present study shows that enrollees who have multiple mental health conditions are at greater risk of developing confusion or memory loss. Clinicians treating patients with mental health conditions should be cognizant of potential cognitive impairment and should educate their patients about this possible outcome associated with disaster-related mental health conditions. In addition, clinicians might recommend that enrollees engage in activities that increase cognitive reserve (e.g., engaging in social activities, physical exercise, and cognitively challenging activities), and that enrollees make dietary and other behavioral changes (e.g., prioritizing high quality sleep) consistent with a healthy lifestyle.

## Figures and Tables

**Figure 1 ijerph-17-07330-f001:**
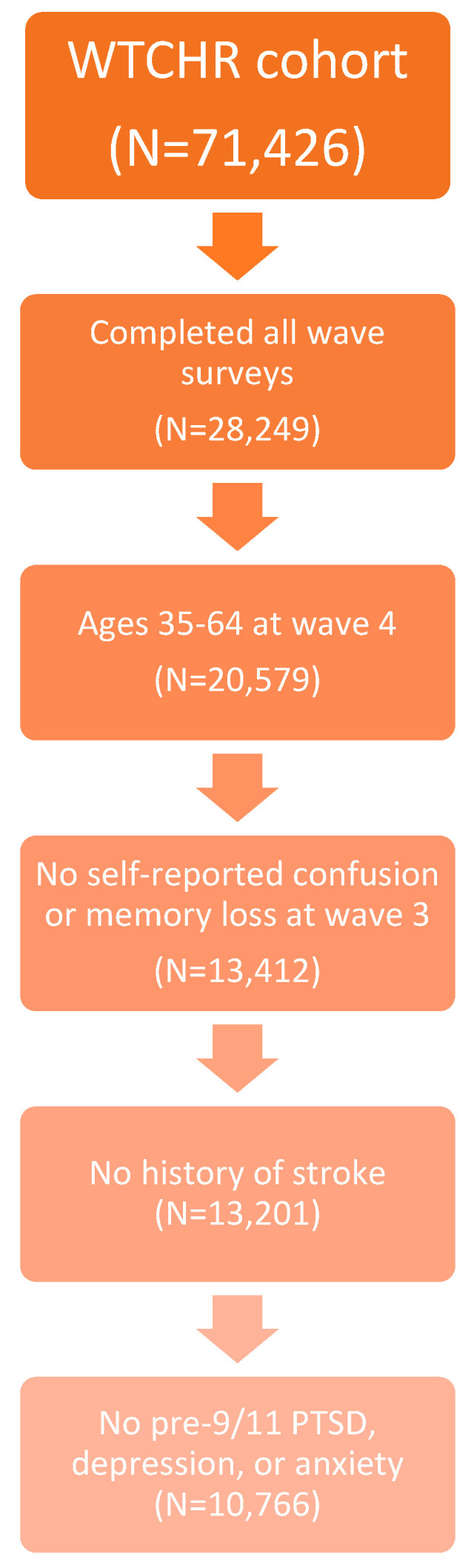
Development of the study sample.

**Table 1 ijerph-17-07330-t001:** Description of study sample, including mental health comorbidity.

Characteristic	Total	
	*n*	%
Total Sample	10,766	100.0
Confusion or Memory Loss		
No	8481	79.73
Yes	2156	20.27
Mental Health Comorbidity at Wave 3		
0 conditions	9637	89.85
1 conditions	609	5.68
2 conditions	292	2.72
3 conditions	188	1.75
Gender		
Male	7007	65.08
Female	3759	34.92
Age at Wave 3		
25–44	3405	31.63
45–64	7361	68.37
Race		
Non-Hispanic white	7936	73.71
Non-Hispanic black	976	9.07
Hispanic	1046	9.72
Non-Hispanic Asian	544	5.05
Non-Hispanic multiracial/other	264	2.45
Education		
High school diploma or less	1851	17.26
Some college	2609	24.33
College or post-graduate degree	6265	58.41
Marital Status at Wave 3		
Married/Living with partner	8016	74.76
Divorced/Separated/Widowed/Never married	2707	25.24
Social Support at Wave 3		
Low	3527	33.48
High	7008	66.52
RRW Status		
Yes	5354	50.33
No	5283	49.67

**Table 2 ijerph-17-07330-t002:** The association of confusion or memory loss (CML) with mental health comorbidity for the full sample ^1,2^.

Characteristic	Confusion or Memory Loss		
	RR	CI (L)	CI (U)
Mental Health Comorbidity at Wave 3			
0 conditions	Ref		
1 conditions	1.85	1.65	2.09
2 conditions	2.13	1.85	2.45
3 conditions	2.51	2.17	2.91

^1^ This model controlled for gender, race/ethnicity, age at wave 3, education, marital status, and social support, and RRW status. ^2^ CI (L) = Lower limit of confidence interval, CI (U) = Upper limit of confidence interval.

**Table 3 ijerph-17-07330-t003:** Risk of worsening Confusion or Memory Loss (CML) based on mental health comorbidity among those who reported any CML ^1^.

Characteristic	Worsening Confusion or Memory Loss		
	RR	CI (L)	CI (U)
Mental Health Comorbidity at Wave 3			
0 conditions	Ref		
1 conditions	1.24	1.01	1.52
2 conditions	1.89	1.57	2.28
3 conditions	1.97	1.62	2.41

^1^ This model controlled for gender, race/ethnicity, age at wave 3, education, marital status, and social support.

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
