# Peer review of "Post-9/11 Mental Health Comorbidity Predicts Self-Reported Confusion or Memory Loss in World Trade Center Health Registry Enrollees"

_ijerph, 2020, doi:10.3390/ijerph17197330_

Round 1
Reviewer 1 Report
important addition to the literature and a nice extension of the works by Clouston and by Singh, both of which only dealt with cognitive decline in 9/11 responders. This current study includes survivors (residents, commuters, nearby workers) as well as responders and that is what is most appealing to me. I also find the inclusion and exclusion criteria quite appropriate. In fact, i believe in the discussion section that you should stress that unlike the 9/11 studies by Clouston and Singh, you have a baseline in wave 3 and therefore included only those who initially did NOT report cognitive issues
I believe the following issues must be addressed.
Intro - too long. The 2nd and 3rd paragraph include non-WTC studies. These should be summarized in 1-2 sentences and these sentences should clearly state that this is non-WTC studies as that was confusing to me until i read the 4th paragraph. Any further details on the non-WTC studies that you feel are relevant should then be moved to the discussion.
Methods - an important covariate that is missing is Responder vs. Survivor. This should be a covariate or a stratified analysis or both. One of the major additions to the literature that result from this study is that the 2 prior studies did not include survivors. You must address this by adding this analysis. You have the data.
Results and Discussion - I have no concerns about your analysis when you correlate a single mental health condition with mild cognitive dysfunction. I have serious concerns about summing the mental health conditions. There is considerable overlap between survey questions in PCL-17 (PTSD) and PHQ (depression). There are also numerous studies showing that there is diagnostic overlap in this population between PTSD and anxiety. At the very least this needs to be addressed as a serious limitation
Discussion - you start by stating you "presume to be mild to moderate confusion or memory loss. You have no basis for presuming mild, moderate or even severe. Only that it is self-reported to be present and to have gotten worse or more frequent.
Also you need to include that your 2 questions on cognitive dysfunction are similar to the CFI (Singh study) but have they been validated as 2 question only survey.
You need to be clear that this does not provide info on how severe it it is or its functional impact. Question 2 only asks if it has occurred more frequently or gotten worse but provides no quantitative assessment. It is self-reported but does not contain a Likert scale for how severe.
Author Response
Manuscript ijerph-950788: Post-9/11 Mental Health Comorbidity Predicts Self-Reported Confusion or Memory Loss in World Trade Center Health Registry Enrollees
|
|
Reviewer’s comments |
Response |
Page (original IJERPH version) |
|
|
Reviewer #1 |
|
|
|
1 |
Intro - too long. The 2nd and 3rd paragraph include non-WTC studies. These should be summarized in 1-2 sentences and these sentences should clearly state that this is non-WTC studies as that was confusing to me until i read the 4th paragraph. Any further details on the non-WTC studies that you feel are relevant should then be moved to the discussion.
|
We have shortened paragraphs 2 and 3 of the introduction and indicated where studies cited are non-9/11-related. |
38-56 |
|
2 |
Methods - an important covariate that is missing is Responder vs. Survivor. This should be a covariate or a stratified analysis or both. One of the major additions to the literature that result from this study is that the 2 prior studies did not include survivors. You must address this by adding this analysis. You have the data. |
We have included Responder status and described the corresponding new statistical analyses, in the methods section. We also corrected the regression results, where necessary, in the abstract and results sections |
123-152 |
|
3 |
Results and Discussion - I have no concerns about your analysis when you correlate a single mental health condition with mild cognitive dysfunction. I have serious concerns about summing the mental health conditions. There is considerable overlap between survey questions in PCL-17 (PTSD) and PHQ (depression). There are also numerous studies showing that there is diagnostic overlap in this population between PTSD and anxiety. At the very least this needs to be addressed as a serious limitation |
We have added text in the discussion concerning the limitations involved in summing the three mental health conditions to create a measure of mental health comorbidity. |
238-242 |
|
4 |
Discussion - you start by stating you "presume to be mild to moderate confusion or memory loss. You have no basis for presuming mild, moderate or even severe. Only that it is self-reported to be present and to have gotten worse or more frequent.
Also you need to include that your 2 questions on cognitive dysfunction are similar to the CFI (Singh study) but have they been validated as 2 question only survey.
You need to be clear that this does not provide info on how severe it it is or its functional impact. Question 2 only asks if it has occurred more frequently or gotten worse but provides no quantitative assessment. It is self-reported but does not contain a Likert scale for how severe. |
We made two modifications. First, we removed the phrase “mild to moderate” from the text, where it referred to the outcome measure used in our study. Other studies we cite sometimes explicitly refer to Mild Cognitive Impairment (MCI), and in those cases we did not remove the mild or moderate descriptions. The second modification was to add material to the limitations section stating that our main outcome measure could not detect the severity of confusion or memory loss. |
13, 41, 182 |
Reviewer 2 Report
Thank you for this valuable opportunity to review this manuscript. The author focuses on the special group experienced the 911 event, focus on the Health Comorbidityon the Self - Reported Confusion or Memory Loss. Generally speaking, the research topic is quite interesting and the data are quite unique. Clinicians treating patients with mental health conditions should be aware of potential cognitive impairment. However, there are still some deficiencies in the study. The specific opinions are as follows:
(1) How do you determine if the respondent's CML is caused by 9/11 memory and not by something else? That is, how did the researchers isolate the effects of other factors on CML.
(2) Problems with the use of data time frames. The author actually has a panel data of issue 4, while the result variable is based on Issue 4 data and the Exposure data is based on Issue 3 data. Why is this arranged?
Author Response
Manuscript ijerph-950788: Post-9/11 Mental Health Comorbidity Predicts Self-Reported Confusion or Memory Loss in World Trade Center Health Registry Enrollees
|
|
Reviewer’s comments |
Response |
Page (original IJERPH version) |
|
|
Reviewer #2 |
|
|
|
1 |
(1) How do you determine if the respondent's CML is caused by 9/11 memory and not by something else? That is, how did the researchers isolate the effects of other factors on CML.
|
Since this study is observational, there is no definitive way to establish causality for the effect of 9/11-related mental health conditions on confusion or memory loss. However, we did exclude all enrollees with pre-9/11 PTSD, depression, and anxiety, and we included a rich set of covariates to account for confounding |
|
|
2 |
Problems with the use of data time frames. The author actually has a panel data of issue 4, while the result variable is based on Issue 4 data and the Exposure data is based on Issue 3 data. Why is this arranged?
|
Although the current study is not causal, we wanted to establish a temporal sequence between mental health comorbidity (at wave 3) and confusion or memory loss (at wave 4), so the former would precede the latter. |
|
Round 2
Reviewer 1 Report
revisions are appropriate and acceptable
thank you
Reviewer 2 Report
The author answered the question I raised, I have no other comments, and suggested to be accepted at current form.